# Neuroprotection of *N*-benzyl Eicosapentaenamide in Neonatal Mice Following Hypoxic–Ischemic Brain Injury

**DOI:** 10.3390/molecules26113108

**Published:** 2021-05-22

**Authors:** Mengya Jiao, Qun Dong, Yiting Zhang, Min Lin, Wan Zhou, Tao Liu, Baohong Yuan, Hui Yin

**Affiliations:** 1Guangdong Provincial Key Laboratory of Pharmaceutical Bioactive Substances, School of Biosciences and Biopharmaceutics, Guangdong Pharmaceutical University, Guangzhou 510006, China; d202081498@hust.edu.cn (M.J.); dq17862812970@163.com (Q.D.); Yeaktine@126.com (Y.Z.); d201980517@hust.edu.cn (W.Z.); yuanbaohong@gdpu.edu.cn (B.Y.); 2School of Pharmacy, Guangdong Pharmaceutical University, Guangzhou 510006, China; 3Department of Microbiology and Immunology, Guangdong Pharmaceutical University, Guangzhou 510006, China; 4School of Clinical Medicine, Guangdong Pharmaceutical University, Guangzhou 510310, China; iaaed@163.com

**Keywords:** maca, *N*-benzyl eicosapentaenamide, neonatal hypoxic–ischemic encephalopathy, neuroprotection, PUMA

## Abstract

Maca (*Lepidium meyenii*) has emerged as a popular functional plant food because of its medicinal properties and nutritional value. Macamides, as the exclusively active ingredients found in maca, are a unique series of non-polar, long-chain fatty acid *N*-benzylamides with multiple bioactivities such as antifatigue characteristics and improving reproductive health. In this study, a new kind of macamide, *N*-benzyl eicosapentaenamide (NB-EPA), was identified from maca. We further explore its potential neuroprotective role in hypoxic–ischemic brain injury. Our findings indicated that treatment with biosynthesized NB-EPA significantly alleviates the size of cerebral infarction and improves neurobehavioral disorders after hypoxic–ischemic brain damage in neonatal mice. NB-EPA inhibited the apoptosis of neuronal cells after ischemic challenge. NB-EPA improved neuronal cell survival and proliferation through the activation of phosphorylated AKT signaling. Of note, the protective property of NB-EPA against ischemic neuronal damage was dependent on suppression of the p53–PUMA pathway. Taken together, these findings suggest that NB-EPA may represent a new neuroprotectant for newborns with hypoxic–ischemic encephalopathy.

## 1. Introduction

Neonatal hypoxic–ischemic encephalopathy (HIE) is a common cause of brain injury due to oxygen deprivation and blood flow reduction [1,2]. HIE is extremely adverse to the development of the brain, which is the main factor causing neurological dysfunction in children [3]. The most common consequences of HIE include cerebral palsy, severe cognitive impairment, and motor and behavioral deficits, resulting in millions of neonatal deaths or long-term disabilities every year [4]. Therapeutic hypothermia is the clinically common treatment for neonatal HIE patients and has been shown to improve neurologic outcomes in survivors [5]. However, just using hypothermia treatment is not sufficient to reduce mortality or avoid severe neurodevelopmental disorders in severe HIE [6]. Despite significant advances in modern medical technology, there is no effective medical treatment for nerve damage caused by HIE [7]. Neuroprotective therapy is of great significance for the long-term recovery of brain function [8]. Therefore, it is necessary to develop safe and effective neuroprotective therapeutics.

Maca (*Lepidium meyenii*) is an annual or biennial herb of the South American cruciferous family, which was classified as a new resource food in 2011 [9,10]. A variety of components with pharmacological and nutritional effects have been identified in maca, which include polysaccharides, phytosterols, alkaloids, glucosinolates, macaenes, and macamides [11,12]. Among these bioactive ingredients, macamides, a group of non-polar, long-chain fatty acid N-benzylamide compounds, were recognized as the characteristic constituents while contributing to the major efficacies in maca such as antifatigue, antiosteoporosis, and improving fertility [13,14,15]. Macamides and their synthetic analogues have recently been reported to display distinct fatty acid amide hydrolase (FAAH) inhibitory activity, which implies that these compounds possess potential neuroprotective and anti-inflammatory activities [16,17].

In the present study, a novel kind of macamide N-benzyl eicosapentaenamide (NB-EPA) has been identified from maca. We further explore its biological activities on hypoxic–ischemic brain damage. Our findings showed that NB-EPA significantly ameliorates HI-mediated neonatal brain injury through the improvement of brain infarction and behavioral disorders. Notably, the neuroprotective effect of NB-EPA on ischemic neuron survival was dependent on suppression of the p53–PUMA signaling pathway.

## 2. Results

### 2.1. Identification and Synthesis of N-benzyl Eicosapentaenamide

Macamides are a class of amide alkaloids formed by benzylamine and a fatty acid moiety, which are considered to be the characteristic marker compounds of maca. Here, we identified a new macamide, NB-EPA (molecular formula C_27_H_37_NO), from maca. As shown in Figure 1A, high-performance liquid chromatography with electrospray ionization mass spectrometry (HPLC-ESI-MS/MS) displayed two peaks which were recorded in the total ion chromatogram of *m*/*z* 392.3 with retention times of 7.0 min and 9.0 min, respectively. The MS/MS spectrum demonstrated that the fragment information of the peak with retention time of 7.0 min was consistent with theoretical fragments of NB-EPA (Figure 1B). The main fragment ion peak detected from this new macamide was *m*/*z* 91.1, corresponding to the benzyl [C_7_H_7_]^+^. The MS/MS fragment ion peaks on the fatty acid side were consistent with the reported MS/MS data of eicosapentaenoic acid in the National Institute of Standards and Technology (NIST) Chemistry WebBook (SRD 69).

Due to the low content of NB-EPA in maca, we then synthesized NB-EPA using the carbodiimide condensation method. The synthesized mixture was purified by a semi-preparative HPLC system. The fraction with retention time from 23.5 to 25.5 min was collected, and the purity of the collected fraction reached 98.2% (Figure 1C,D). HPLC-MS/MS analysis demonstrated that biosynthesized NB-EPA and natural NB-EPA had identical chromatographic separation characteristics and secondary mass spectrometry fragment information (Figure 1E–G). Together, these data suggest that a new macamide NB-EPA was identified from maca and successfully synthesized.

### 2.2. NB-EPA Reduces Infarct Volumes and Neurobehavioral Deficits in Neonatal HI Brain Injury

We then investigated the bioactivity of NB-EPA in neonatal mice following hypoxic–ischemic (HI) brain injury. Images of brain appearance showed that ischemic brain tissue became white and swollen in the vehicle group on days 1, 3, 7 and 14 post-HI (Figure 2A). By contrast, administration of NB-EPA at a dose of 250 μg/day for 3 d significantly alleviated brain swelling and brain water content (Figure 2A,B). Staining with 2,3,5-triphenyltetrazolium chloride (TTC) showed that NB-EPA treatment (11.13 ± 2.58%) clearly reduced the size of cerebral infarction 3 d post-HI compared with the vehicle (37.01 ± 4.68%) (Figure 2C). Neurobehavioral tests demonstrated that mice in the vehicle group displayed significant neurobehavioral disorders at 1, 3, and 7 d post-HI (Figure 2D–F). On the contrary, NB-EPA treatment clearly improved these neurological defects (Figure 2D–F). Thus, these data indicate that NB-EPA protects against ischemic lesion and neurobehavioral deficits in neonatal mice after HI brain injury.

### 2.3. NB-EPA Decreases HI-Induced Brain Damage by Suppression of Neuronal Apoptosis

Next, we explored whether the protection of NB-EPA in HI brain injury was implicated in the modulation of neuronal death. Hematoxylin–eosin (HE) staining showed that neuron arrangements in the cortex, hippocampus dentate gyrus (DG), cornus ammonis (CA) 1 and CA3 regions were disordered. and the nerve fibers were loosened and vacuolated in ischemic brains after HI injury. However, treatment with NB-EPA clearly abrogated neuronal cell damage (Figure 3A). Western blot confirmed that NB-EPA markedly reduced the expression of pro-apoptotic proteins such as p53, PUMA and Bax in ischemic brain tissue (Figure 3B,C). In addition, neuronal nuclei antigen (NeuN) and TUNEL double staining demonstrated a significant decrease in the number of TUNEL^+^NeuN^+^ cells 7 d post-HI in NB-EPA-treated mice (5.80 ± 1.35%) compared with vehicles (14.39 ± 2.55%) (Figure 3D). Together, these data imply that the beneficial effect of NB-EPA in HI brain insults is involved in the inhibition of neuronal cell apoptosis.

### 2.4. NB-EPA Decreases HI-Induced Brain Damage by the Suppression of Neuronal Apoptosis

Nissl staining showed that a large number of neurons exhibited atrophy, swelling, and nuclear pyknosis, and even died with the disappearance of Nissl bodies in the ischemic brain. By contrast, NB-EPA apparently improved the neuronal survival and neuron rearrangement after HI injury (Figure 4A,B). Western blot demonstrated that NB-EPA elevated the levels of phosphorylated AKT (p-AKT), acting as a pro-survival signaling, in ischemic brain tissue 7 d after HI challenge (Figure 4C,D). Doublecortin (DCX) is a marker for neuronal precursors and young migrating neurons [18], and Ki-67 is a marker of cell proliferation [19]. NB-EPA treatment (19.83 ± 1.25 cells/mm^3^) remarkably increased the number of Ki-67^+^DCX^+^ cells 7 d post-HI compared with the vehicle (11.83 ± 1.08 cells/mm^3^) (Figure 4E). Therefore, these data imply that NB-EPA facilitates neuronal survival and proliferation in neonatal mice after HI brain injury.

### 2.5. NB-EPA Decreases HI-Induced Brain Damage by the Suppression of Neuronal Apoptosis

Based on the above protective effect of NB-EPA in neonatal HI brain damage, we further assessed the direct influence of NB-EPA on neuronal survival in vitro. Primary cortical neurons cultured for 7 days in vitro were round and small, with rich synapses and interconnected networks (Figure 5A). The purity of neurons was determined by flow cytometry, and the positive rate was over 95% (Figure 5B). NeuN^+^ neurons were also identified by immunofluorescence (Figure 5C). These results show that primary cortical neurons were successfully cultured.

Then, the neurons were subjected to 3 h oxygen–glucose deprivation (OGD) and treated with different concentrations of NB-EPA. Hypoxic neurons appeared with atrophy and nuclear pyknosis in vehicle PBS-treated cultures. In contrast, NB-EPA treatment significantly improved neuronal survival after OGD (Figure 5D). Neuron survival was measured 24 h later using cell counting kit-8 (CCK-8) assay. As shown in Figure 5E, NB-EPA treatment apparently rescued the survival rate of neurons after OGD injury. The maximal efficacy of NB-EPA was achieved at 1 μM, which was used for all following in vitro experiments. Flow cytometry analysis confirmed that the number of annexin-positive neurons in NB-EPA-treated cultures was significantly decreased compared with vehicle PBS-treated cultures (Figure 5F). Meanwhile, NB-EPA treatment also markedly inhibited the expression of apoptosis-related proteins, including p53, PUMA, and Bax in OGD-conditioned neurons (Figure 5G,H). Therefore, these data imply that NB-EPA attenuates neuronal damage through the upregulation of neuronal survival while inhibiting neuronal apoptosis.

### 2.6. NB-EPA Protects Neurons from Apoptosis by the Suppression of p53-PUMA Signaling

The p53–PUMA pathway is known to participate in ischemia/reperfusion-induced cerebral neuronal cell apoptosis [20]. We explored here whether the protection of NB-EPA against neuronal death is related to inhibition of the p53–PUMA pathway. As shown in Figure 6A, NB-EPA abrogated OGD-conditioned cell apoptosis, but had no effect on samples pretreated with Pifithrin-α (PFTα), a p53 inhibitor. Meanwhile, NB-EPA treatment markedly inhibited the expression of pro-apoptotic protein Bax in OGD-conditioned neurons but had no effect on those pretreated with PFTα (Figure 6B). PUMA expression enhanced noticeably in neurons after OGD challenge, whereas PUMA mRNA and protein expression did not change in PFTα-pretreated neurons, indicating that PUMA induction in neurons is p53-dependent (Figure 6C). NB-EPA treatment inhibited OGD-induced PUMA expression in neurons but had no effect on PFTα-pretreated neurons (Figure 6C). Cerebral neurons with specific PUMAα silencing showed a lower percentage of apoptotic cells (Figure 6D) and a reduced expression of Bax protein (Figure 6E) at 24 h after OGD challenge compared to PUMA-expressing neurons. Therefore, these data indicate that NB-EPA-mediated inhibition of OGD-induced neuron apoptosis is largely dependent on repression of the p53–PUMA pathway.

## 3. Discussion

Neonatal hypoxic–ischemic damage caused by perinatal asphyxia is one of the most common diseases in the neonatal period. There are a lack of effective pharmaceutical therapeutic interventions that reduce cerebral injury or improve neurological function in infants [21]. In the present study, we found that treatment with a new macamide NB-EPA significantly alleviated hypoxic–ischemic brain injury in neonatal mice. The neuroprotection of NB-EPA was related to the upregulation of neuronal survival while inhibiting neuronal death. Notably, NB-EPA-mediated antiapoptotic mechanism was dependent on suppression of the p53-PUMA signaling.

Macamides, known as the characteristic marker compounds of maca, consist of benzylamine and a long chain fatty acid moiety with a variable degree of unsaturation [22]. Twenty-three kinds of macamides have been reported in maca extract, among which *N*-benzylhexadecanamide is the most abundant compound in maca from Peru, while *N*-benzyl-9Z,12Z-octadecadienamide is the richest compound in the Yunnan maca [23]. In this study, we identified a novel macamide NB-EPA in maca from Yunnan. The macamide NB-EPA is composed of benzylamine and eicosapentaenoic acid (EPA; 20:5, n-3), one of the main components of omega-3 polyunsaturated fatty acid (n-3 PUFA). n-3 PUFAs and their metabolites have been reported to exist in the central nervous system and play crucial roles in brain function and disease, such as neurotransmission, neurogenesis, and neuroinflammation [24]. Of note, some studies have suggested that EPA, rather than n-3 PUFAs and docosahexaenoic acid (DHA; 22:6, n-3), is related to lower risks of most types of ischemic stroke [25]. In agreement with these findings, we demonstrated that this novel macamide NB-EPA not only alleviates brain injury, but also improves neurobehavioral deficits after HI injury.

Neurons are the most basic structural and functional units of the nervous system, which have the function of connecting and integrating input information and sending out information. When the brain is injured, caused by hypoxia and ischemia, a large number of neurons are damaged, which, in turn, causes neurological dysfunction. p53, a key regulator of cellular stress response, can be activated in the ischemic areas after brain injury [26]. It can promote neuronal apoptosis, and p53 deficiency or the application of p53 inhibitors can significantly attenuate brain damage in various stroke models [27]. p53-mediated apoptosis occurs through a variety of molecular mechanisms, among which PUMA, a p53-upregulated modulator of apoptosis, is a potent proapoptotic gene downstream of p53 [28]. Studies have shown that the p53–PUMA pathway is involved in mitochondrial inhibitor-induced apoptosis of striatum neurons in rats [29]. Inhibition of p53–PUMA feedback loop activation by p53 inhibitors and PUMA siRNA can reduce neuronal apoptosis and inflammation induced by ischemia–reperfusion [30]. Here, our findings are consistent with these studies showing that PUMA is a strong executor of p53-mediated apoptosis in neurons after HI brain injury. NB-EPA treatment inhibits HI-induced elevation of PUMA in neurons. Furthermore, the protective effects of NB-EPA on neuronal survival are dependent largely on the repression of PUMA signaling.

Incremental evidence shows that n-3 PUFAs contribute to neuron survival and neuranagenesis after ischemic cerebral damage [31,32]. They can improve the prognosis of stroke and limit further neuron damage [33]. Although ischemic stroke induces the proliferation and differentiation of neural progenitor cells in the subventricular zone (SVZ), most of the newly generated neurons died shortly after stroke [34]. n-3 PUFAs have been reported not only to enhance the survival of immature neurons, but also to facilitate their maturation in cortical parenchyma after stroke damage [35]. Recently, several studies also suggested the beneficial effect of macamides in the amelioration of neuronal damage, which is related to the improvement of mitochondrial respiratory function [36,37]. The AKT signal transduction pathway plays an important role in the anti-apoptosis mechanism. When the brain is injured, the AKT pathway is activated, and the body initiates self-protection and injury repair, leading to increased expressions of p-AKT [38,39,40,41]. In line with these findings, we observe that treatment with NB-EPA protects neuronal survival against HI-mediated brain injury through the activation of pro-survival AKT signaling while inhibiting the pro-apoptotic pathway.

For a long time, neurobiologists believed that neural stem cells (NSCs) disappeared shortly before or after birth, and neurogenesis stopped at that time. The notion that there are no new neurons in the mature brain began to change in the 1960s. With the development of research, it was found that adult NSCs mainly exist in the SVZ and the subgranular zone (SGZ) of the hippocampal dentate gyrus [42,43,44]. Furthermore, it was later confirmed that NSCs also exist in other extensive areas of the central nervous system (CNS) [45]. Recent studies have shown that endogenous NSCs in the cortex can be activated after brain injury to contribute to the repair of hypoxic–ischemic brain injury through self-renewal, proliferation, and the generation of new neurons, astrocytes, and oligodendrocytes [46,47]. DCX contributes to neuronal repair by stabilizing microtubules in neuronal cells and represents a marker for tracking the migration of new neurons into the injured sites of the brain [48]. Hypoxic–ischemic brain injury and severe traumatic brain injury increase the production of new striatum neurons that express DCX [49]. Ki-67 is a nuclear antigen associated with proliferating cells, covering every proliferation cycle other than the G0 phase. In this study, NB-EPA administration apparently promotes neurogenesis, which is associated with the upregulation of Ki-67^+^DCX^+^ cells after HI brain injury. However, whether the increase in the number of Ki-67^+^DCX^+^ cells is related to the proliferation and differentiation of neural stem cells induced by NB-EPA still needs to be further studied.

## 4. Materials and Methods

### 4.1. Extraction of NB-EPA from Maca

The macamide NB-EPA was extracted from maca as described previously [50]. Briefly, diethyl ether solution (30 mL) and dried maca powder (1 g; Taike Biotechnology, Xi’an, China) were mixed and ultrasonically extracted at room temperature for 30 min. After centrifugation at 4000× *g* for 15 min, the supernatants were collected and evaporated to dryness using a rotary vacuum evaporator. Then, the dried extracts were treated by the acid–base method and enriched by *n*-hexane extraction. The extracts containing macamides were prepared for HPLC-MS/MS analysis.

### 4.2. Synthesis and Purification of NB-EPA

Microalgae oil was transformed into free fatty acid mixtures by a transesterification reaction as described previously [51]. Briefly, a NaOH–CH_3_OH solution (0.4 M) and microalgae oil (10:1; Taike Biotechnology, Xi’an, China) were mixed and incubated with continuous stirring at 50 °C for 60 min. Then, the reaction mixture was transferred into a separating funnel, and equal amounts of n-hexane and water were added with continuous stirring for 30 min. Subsequently, the water layer was collected, and equal amounts of hydrochloric acid (3 M) and n-hexane were added and stirred under ultrasonic conditions for 10 min. After washed by sodium hydroxide (0.125 M) and hydrochloric acid (0.29 M), the n-hexane layer was collected and evaporated to dryness for NB-EPA synthesis.

A dichloromethane solution (50 mL) containing HOBt·H_2_O (103 mg), EDAC (146 mg), triethylamine (264 μL), and free fatty acid (264 μL) was stirred at room temperature for 20 h. Benzylamine (83 μL; Aladdin Reagents, Shanghai, China) was then added to the reaction mixture. After 4 h, the final solution was evaporated to dryness. Subsequently, equal amounts of hydrochloric acid (0.29 M) and n-hexane were added and stirred under ultrasonic conditions for 10 min. After washing with sodium hydroxide (0.125 M) and hydrochloric acid (0.29 M), the n-hexane layer was collected and evaporated to dryness for semi-preparation HPLC purification (Waters, Milford, MA, USA).

### 4.3. HPLC MS/MS Analysis

The HPLC MS/MS analysis was carried out on Agilent 1100 series liquid chromatography apparatus combined with electrospray ionization mass spectrometry (LCQ DecaXP, Thermo electron, San Jose, CA, USA), which was equipped with an Agilent Zorbax SB C18 column (2.1 × 150 mm, 5 µm). The isocratic elution system comprised acetonitrile (90%) and water (10%) in 15 min. The flow rate was 0.2 mL/min. The outlet of the column was introduced into the ion source of mass spectrometer. The spray voltage was set to 4.5 kV, and the heat capillary was kept at 300 °C. The MS scan was set as single ion monitoring model with *m*/*z* 392.3. The collision energy value was set as 40%. MS data acquisition and process were performed with Xcalibur 1.3 software (Thermo Fisher, Waltham, MA, USA).

### 4.4. Animals and Treatment

Seven-day-old (P7) C57BL/6 mouse pups were obtained from the Center of Experimental Animals of Guangdong Province and raised under specific pathogen-free conditions. One male and two females were mated for reproduction, and their pups at seven days old (P7) were chosen for the following experiments. All animal treatments were strictly in accordance with the National Institutes of Health Guide for the Care and Use of Laboratory Animals (NIH Publication 85–23, 1996), and were approved by the Institutional Ethical Committee of Guangdong Pharmaceutical University (Approval Number: gdpulac2017083). Mice were injected i.p. with NB-EPA (250 μg per mouse) or vehicle PBS (control) for three days after hypoxic–ischemic (HI) operation. Remarkably, the first injection was administered at the time of HI operation.

### 4.5. Neonatal Hypoxia–Ischemia Model

The modified Rice–Vannucci model was used to establish the HIE mouse model [52]. P7 C57BL/6 mouse pups weighing 3.5 to 4.5 g were anesthetized by inhalation of isoflurane and treated with bipolar electrical coagulation (Vetroson) for unilateral left common carotid artery occlusion. The incision was cemented with a tissue adhesive (3M Vetbond). After a recovery period of 2.5 h, pups were placed in a hypoxia chamber (containing 8% oxygen in a balance with 92% nitrogen) maintained at 37 °C for 90 min. Pups were then recovered for 0.5 h and returned to their dams. For pups in the sham group, a small incision was made in their neck without artery occlusion, and they were placed in a chamber at normal air temperature.

### 4.6. Brain Water Content Measurement

Water content of ischemic brain tissue was determined by the dry–wet method. Three days after the termination of hypoxic insult, brains of neonatal mice in every group were taken out. The olfaction bulb and cerebellum were removed after rinsing with PBS, and the cerebral hemispheres were separated. The cerebral hemisphere on the ischemic side was accurately weighed and measured as wet weight (WW). Then, the brain tissue was placed in a constant temperature drying oven at 80 °C for 72 h, and the brain tissue was accurately measured as dry weight (DW). Brain water content is calculated according to the following formula: brain water content (%) = (WW − DW)/WW × 100%.

### 4.7. Infarct Volume Measurement

Three days after the termination of hypoxic insult, brains of pups in every group were removed and sectioned coronally into 2 mm slices. Brain slices were incubated in 2% TTC in PBS for 15 min at 37 °C, and then transferred to 4% paraformaldehyde (PFA) solution for fixation for about 24 h. The infarct volume was traced and analyzed by ImageJ software (version 1.8.0, National Institutes of Health, Bethesda, MD, USA).

### 4.8. Neurobehavioral Evaluation

Pups in each treatment group were subjected to three neurobehavioral tests 1, 3 and 7 days after HI: (1) geotaxis reflex for diagnosing the function of vestibular and proprioception; (2) cliff avoidance reaction for assessing the ability of rodents to respond to adverse environments; and (3) the grip test for evaluating grip force and fatigability [53]. All the experiments were observed by at least two naïve lab workers, and all animal groups maintained the same testing methods.

### 4.9. Histopathologic Analysis

The sections were deparaffinized in xylene and rehydrated in 100% to 70% gradient ethanol. Then, they were stained with HE or Nissl. The slices were washed with double-distilled water, dehydrated in ethanol, cleaned with xylene, and examined with an Olympus BX51 microscope. ImageJ software (version 1.8.0) was used to calculate the area and number of neurons in the corresponding region including the cerebral cortex, hippocampus DG, CA1 and CA3 regions, and then the neuron density in the region was further calculated. Nissl bodies are large and numerous, which indicates that neurons have strong functions of synthesizing proteins. When neurons are injured, the number of Nissl bodies decreases or even disappears. An expert in the field of pathology, blinded to allocation, assessed the sections for HE or Nissl staining.

### 4.10. Primary Mouse Cortical Neuron Cultures

Primary mouse cortical neurons were cultured as described previously [54]. Briefly, newborn mouse pups were euthanized after being disinfected with 75% ethanol. Cerebral cortices from brain tissue were dissected and cells were dissociated by incubation with OPC papain solution (1.54 mg/mL papain, 360 μg/mL *L*-cysteine and 60 μg/mL DNase I) for 15 min at 37 °C. To obtain a single cell suspension, the cells were filtered by a 40 μm cell strainer. Dissociated cells were planted onto a poly-L-lysine-coated culture plate in neurobasal media containing 2% B27. Cells were cultured in a cell culture incubator (95% O_2_ and 5% CO_2_ at 37 °C) for 7–9 days before experiments.

### 4.11. Lentivirus Transduction

Primary cortical neurons were infected with either PUMAα shRNA lentiviral particles (PUMA shRNA LV, Santa Cruz Biotechnology, Santa Cruz, CA, USA) or control shRNA LV (Santa Cruz Biotechnology, Santa Cruz, CA, USA) in neurobasal media with 5 μg/mL polybrene at multiplicities of infection (MOI) of 15. After 12 h of culturing, the medium was replaced by fresh neurobasal media containing 2% B27 in order to remove debris and inactive lentiviruses.

### 4.12. Oxygen–Glucose Deprivation (OGD) Challenge

The media of neuron cultures were replaced with the glucose-free DMEM and transferred into an anaerobic chamber (5% CO_2_ and 95% N_2_ at 37 °C) for 3 h. The neurons were incubated again in neurobasal medium containing 2% B27 and returned to the normoxic conditions for another 24 h. For NB-EPA treatment, cultured cortical neurons (6 × 10^5^) were treated with different concentrations (0.1, 1, 10 and 100 μM) throughout the whole period of OGD/R, with or without pretreatment of the p53 inhibitor Pifithrin-α (10 μM; Selleck, Santa Cruz, CA, USA). Cells without exposure to OGD/R were defined as the control group. The cells were collected after OGD for the following experiments.

### 4.13. Cell Viability Assay

Cell viability was assessed by CCK-8 (Dojindo, Japan) according to the manufacturer’s instructions. Cells were seeded at a density of 5 × 10^3^ cells per well in 96-well plates. After undergoing OGD, the cells were cultured with NB-EPA for 24 h and subsequently with CCK-8 solution for 2 h at 37 °C. The optical absorbance at 450 nm was detected using a microplate reader (Model 680, Bio-Rad Laboratory, Hercules, CA, USA).

### 4.14. Flow Cytometry

For analysis of the purity of primary neurons, cells were fixed and permeabilized in 0.25% saponin. Cells were stained with a primary anti-NeuN antibody (Sigma-Aldrich, A60, Darmstadt, Germany) followed by a secondary PE anti-mouse IgG1 antibody (Biolegend, San Diego, CA, USA). Cell apoptosis was measured using the Annexin V apoptosis detection kit (eBioscience, 88–8007-72) according to the manufacturer’s directions. For analysis of PUMA protein levels, cells were fixed and permeabilized in 0.25% saponin. Cells were stained with a primary anti-PUMA antibody (MultiSciences Biotech, China, ab37355–100) followed by a secondary anti-rabbit PE antibody (Biolegend). Flow cytometric analysis was performed with a FACSCalibur cytometer (BD Biosciences, Franklin Lakes, NJ, USA) and CellQuest v3.3 software (BD Biosciences, Franklin Lakes, NJ, USA).

### 4.15. Immunofluorescence

Primary neurons were incubated with 10% bovine serum albumin (BSA) for 1 h, and then stained overnight at 4 °C with anti-NeuN (1:50; Sigma-Aldrich, MAB377). Mouse pups were anesthetized by the inhalation of isoflurane and transcardially perfused with 4% PFA in PBS. The brain tissues were fixed with 4% PFA for 24 h, embedded in paraffin, and sliced coronally to 5 μm thick. Tissue sections were dewaxed, and quenched with 3% hydrogen peroxide for 30 min. The sections were incubated with 10% BSA for 1 h, and then stained overnight at 4 °C with either anti-NeuN (1:50; Sigma-Aldrich, MAB377), anti-DCX (1:400; Invitrogen, MA5–17066), or anti Ki-67 (1:500; Abcam, ab15580). On the following day, sections or primary neurons were incubated with PE- or FITC-mouse secondary antibodies (Biolegend) for 1 h. TUNEL staining was performed with the in situ cell death detection kit (Roche). The slides were counterstained with nuclear dye DAPI and observed on an Olympus BX51 microscope.

### 4.16. Western Blot Analysis

Total proteins from ischemic brain tissue and primary neurons were extracted by tissue homogenization in RIPA buffer containing a proteinase inhibitor cocktail (Santa Cruz Biotechnology, Santa Cruz, USA). Equal amounts of lysed and boiled protein (30 μg/well) were separated on 12% SDS-PAGE gel and proteins were transferred onto polyvinylidene difluoride (PVDF) membranes. Blots were probed with anti-Bax (Abcam, ab32503), anti-p53 (MultiSciences Biotech, Hangzhou, China, ab37355–100), anti-PUMA (MultiSciences Biotech, China, ab40081–100), anti-AKT (Cell Signaling Technology, 2920s), and anti-phospho-AKT (Cell Signaling Technology, 4060), and then detected using HRP-conjugated secondary antibody (Abcam). β-actin (Sino Biological Inc, Beijing, China, 100166-MM10) was used as a loading control. The protein bands were visualized with enhanced chemiluminescence reagents, and the resulting images were analyzed using ImageJ and normalized to β-actin.

### 4.17. Real-Time Quantitative PCR

Total RNA from primary neurons was extracted with TRIzol reagent (Invitrogen) and reverse transcription was performed using the first strand cDNA synthesis kit (Invitrogen). The SYBR Green qPCR kit (Invitrogen) was used to detect the expression of target genes PUMA and GAPDH, and the primers used in the PCR amplification were: PUMA forward 5′-CCT CCT TTC TCC GGA GTG TTC A-3′, reverse 5′-ATA CAG CGG AGG GCA TCA GG-3′ and GAPDH forward 5′-TTC ACC ACC ATG GAG AAG GC-3′, reverse 5′-GGC ATG GAC TGT GGT CAT GA-3′. All RT-qPCR reactions were performed with an ABI PRISM^®^ 7000 Sequence Detector Systems (Applied Biosystems, Foster City, CA, USA), and expression levels of genes were normalized to GAPDH mRNA using the comparative threshold cycle (CT) method.

### 4.18. Statistical Analysis

All statistical analyses were performed using SPSS version 18.0 and presented as the mean ± SEM. Statistical differences between groups were evaluated by the Student’s *t*-test or one-way ANOVA. *p*-values less than 0.05 were considered statistically significant.

## 5. Conclusions

In conclusion, our presented data suggest that the new macamide NB-EPA provides a neuroprotective effect on HI-mediated neonatal brain injury. NB-EPA obviously decreased cerebral insults and behavioral deficits. Moreover, the beneficial effect of NB-EPA is related to improvements of neuronal survival by downregulation of the p53–PUMA pathway (Figure 7). Translationally, these findings imply that NB-EPA may represent a new neuroprotectant for newborns with hypoxic–ischemic encephalopathy.

## Figures and Tables

**Figure 1 molecules-26-03108-f001:**
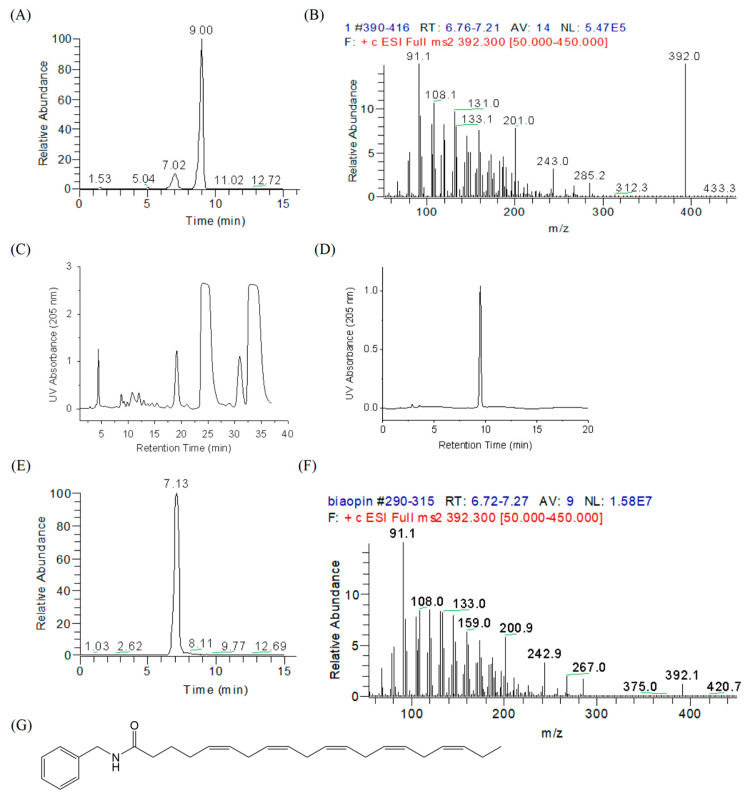
Chromatograms and MS/MS spectra of natural NB-EPA and synthetic NB-EPA. (**A**) Representative total ion current (TIC) chromatogram obtained in positive selected ion monitoring (SIM) mode for a maca extract sample. (**B**) MS/MS spectra of *m*/*z* 392.3 ([NB-EPA^+^H]^+^) for the maca extract sample. (**C**) Semi-preparation HPLC chromatogram of synthetic NB-EPA materials. (**D**) HPLC chromatogram of NB-EPA fractions (23.5–25.5 min) collected from a semi-preparation HPLC system. (**E**) Representative TIC chromatogram obtained in positive SIM mode for a synthetic NB-EPA sample. (**F**) MS/MS spectra of *m*/*z* 392.3 for the biosynthetic NB-EPA sample. (**G**) Chemical structure of NB-EPA.

**Figure 2 molecules-26-03108-f002:**
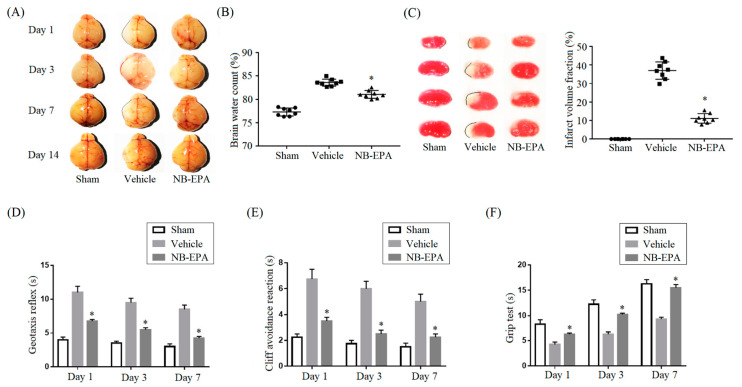
NB-EPA administration reduces cerebral infarction and behavioral defects in neonatal mice after HI brain injury. (**A**) Representative images of brain appearance at 1, 3, 7 and 14 days post-HI. (**B**) Statistical analysis of the brain water content from every group of mice at 3 days post-HI. (**C**) Representative photographs of TTC-stained coronal brain sections from every group of mice at 3 days after HI and quantitative analysis of the infarct volume. (**D**–**F**) Neurobehavioral outcomes of the geotaxis reflex (**D**), cliff avoidance reaction (**E**) and grip test (**F**) at 1, 3 and 7 days post-HI. Data are mean ± standard error of the mean (SEM) (*n* = 8 per group). * *p* < 0.05 compared to vehicle.

**Figure 3 molecules-26-03108-f003:**
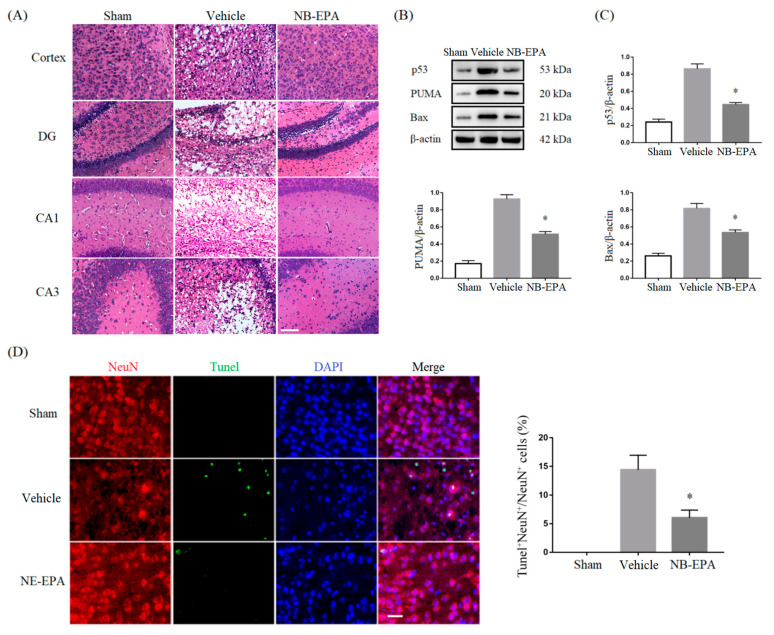
NB-EPA inhibits neuronal death in neonatal mice after HI brain injury. (**A**) Representative photographs of HE staining in the cortex, hippocampus DG, CA1 and CA3 regions from every group of mice at 7 days post-HI (Scale bar, 75 μm). (**B**) Representative Western blot images of p53, PUMA, and Bax from every group of mice at 7 days after HI. (**C**) Densitometric analysis of data is shown in (**B**). (**D**) Representative images of NeuN (red) and TUNEL (green) from every group of mice at 7 days post-HI (left, Scale bar, 25 μm) in cerebral cortex; Quantification of NeuN and TUNEL dual-labeled cells at 7 days post-HI (Right). Data are mean ± SEM (*n* = 8 per group). * *p* < 0.05 compared to vehicle.

**Figure 4 molecules-26-03108-f004:**
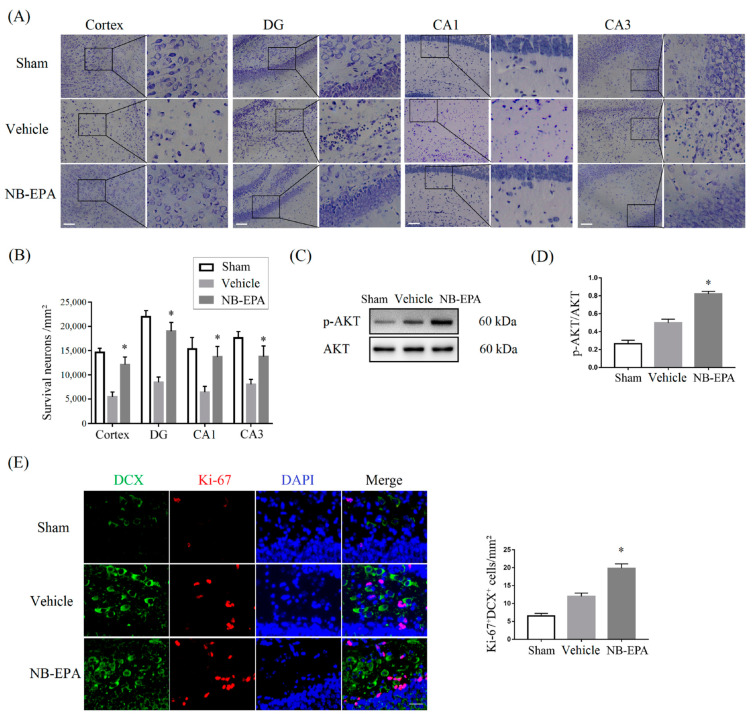
NB-EPA improves the neuronal survival and proliferation in neonatal mice after HI brain injury. (**A**) Representative photographs of Nissl staining in the cortex, hippocampus DG, CA1 and CA3 regions from every group of mice at 7 days post-HI (Scale bar, 75 μm). (**B**) Statistical analysis of the number of neurons is shown in (**A**). (**C**) Representative Western blot images of p-AKT and AKT from every group of mice at 7 days after HI. (**D**) Densitometric analysis of data is shown in (**C**). (**E**) Representative images of DCX (green) and Ki-67 (red) from every group of mice at 7 days after HI (left, Scale bar, 25 μm) in hippocampus DG regions; quantification of DCX and Ki-67 dual-labeled cells at 7 days post-HI (right). Data are mean ± SEM (*n* = 8 per group). * *p* < 0.05 compared to vehicle.

**Figure 5 molecules-26-03108-f005:**
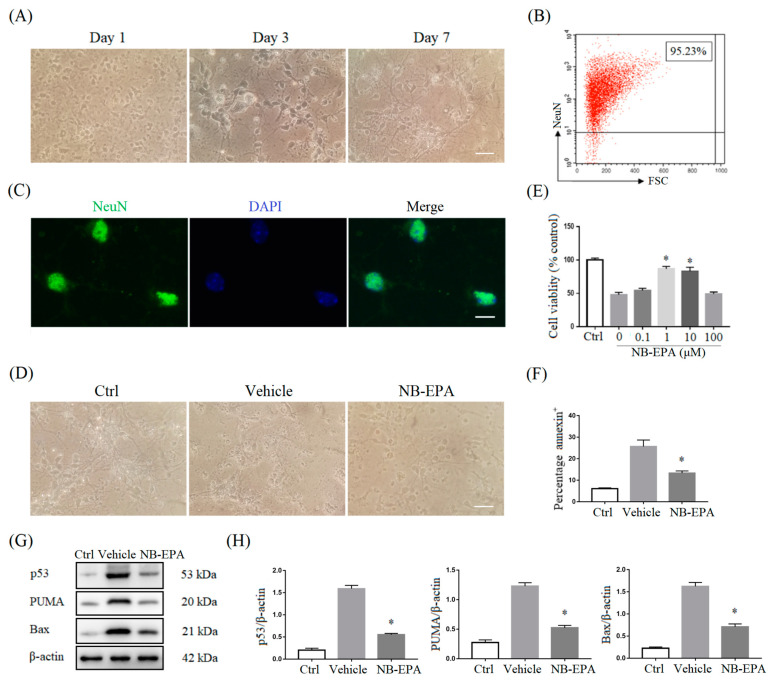
NB-EPA reduces ischemic neuronal injury after OGD challenge. (**A**) Representative images of primary cortical neurons cultured at 1, 3 and 7 days (Scale bar, 25 μm). (**B**) The percentage of NeuN^+^ cells from primary cortical neurons cultured for 7 days in vitro. (**C**) Representative images of NeuN (green) and 4′,6-diamidino-2-phenylindole (DAPI) (blue) from primary cortical neurons cultured for 7 days in vitro (Scale bar, 10 μm). (**D**) Representative images of primary cortical neurons from every group at 24 h of culture after 3 h OGD (Scale bar, 25 μm). (**E**) CCK-8 assay in neuron-enriched cultures subjected to 3 h OGD or control conditions followed by treatment with a range of concentrations of NB-EPA for another 24 h. (**F**) The percentage of annexin^+^ neurons at 24 h of culture after 3 h OGD. (**G**) Representative Western blot images of p53, PUMA, and Bax from every group at 24 h of culture after 3 h OGD. (**H**) Densitometric analysis of data is shown in (**G**). Data are mean ± SEM (*n* = 3 in each group). * *p* < 0.05 compared to vehicle.

**Figure 6 molecules-26-03108-f006:**
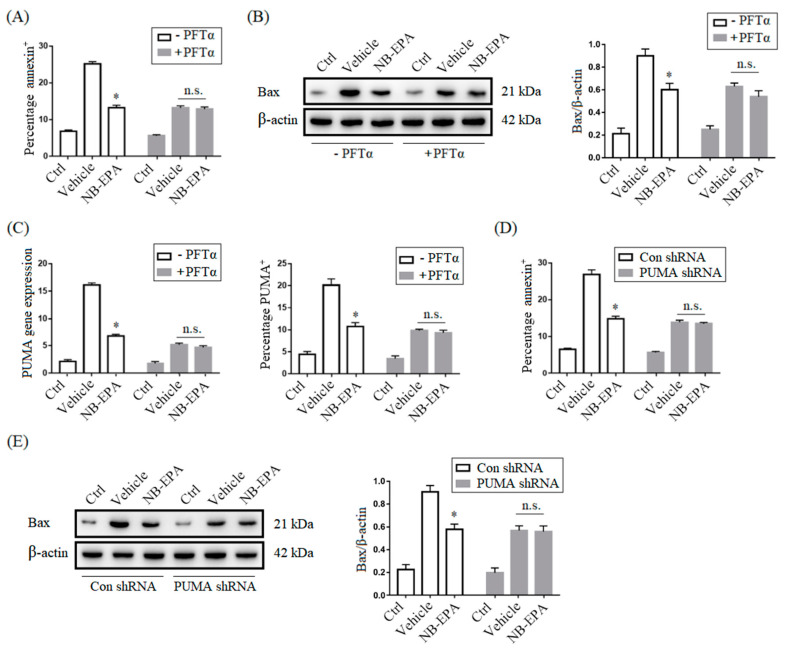
NB-EPA reduces neuron apoptosis through inhibition of the p53–PUMA pathway. (**A**) The percentage of annexin^+^ neurons with or without p53 inhibition at 24 h of culture with or without NB-EPA and after 3 h OGD. (**B**) Representative Western blot images of Bax with or without p53 inhibition at 24 h of culture with or without NB-EPA and after 3 h OGD (left); densitometric analysis of Bax protein expression (right). (**C**) PUMA mRNA expression in neurons at 12 h of culture with or without NB-EPA and after 3 h OGD (left); mean percentages of PUMA protein expression in neurons at 24 h of culture with or without NB-EPA and after 3 h OGD (right). (**D**) The percentage of annexin^+^ neurons with or without PUMA knockdown at 24 h of culture with or without NB-EPA and after 3 h OGD. (**E**) Representative Western blot images of Bax with or without PUMA knockdown at 24 h of culture with or without NB-EPA and after 3 h OGD (left); densitometric analysis of Bax protein expression (right). Data are mean ± SEM (*n* = 3 per group). * *p* < 0.05 compared to vehicle. n.s. not significant.

**Figure 7 molecules-26-03108-f007:**
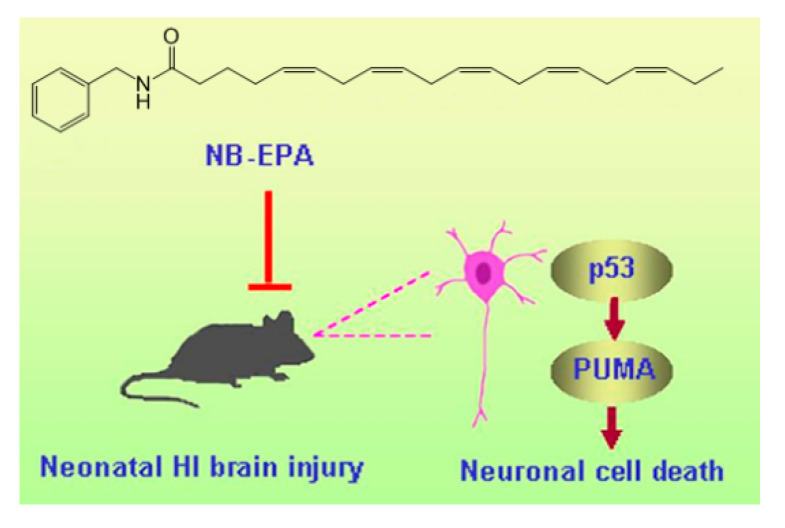
p53, a key regulator of cellular stress response, can be activated in the ischemic areas after brain injury, which then triggers downstream proapoptotic PUMA signaling, thereby resulting in neuronal cell apoptosis, cerebral infarction, and neurobehavioral defects. In contrast, these effects were ameliorated by NB-EPA-mediated suppression of the p53–PUMA pathway.

## Data Availability

The data that support the findings of this study are available from the corresponding author upon reasonable request.

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
