# Peer review of "Neuroprotection of N-benzyl Eicosapentaenamide in Neonatal Mice Following Hypoxic–Ischemic Brain Injury"

_molecules, 2021, doi:10.3390/molecules26113108_

Round 1

Reviewer 1 Report

The authors demonstrated the effectiveness of NB-EPA, a macamide from maca, in hypoxic-ischemic brain damage in newborn mice. The compound was isolated, characterized and successfully synthesized. Its effects on the HI-brain damage was soundly demonstrated by many analyses, both in vitro and in vivo.

I congratulate the authors for the very high quality of the paper.

Please find below a short list of major and minor issues I suggest the author to address in order to improve their paper

MINOR

Line 93: TCC staining, please explain the acronym

Line 110: DG and CA3 regions? Please detail the acronym

Line 132: please explain breafly what Ki-67 and CDX are, so that the reader can understand the results

MAJOR

Figure 2, 3 and 4: the significance is indicated vs untreated controls. Does it mean that NB-EPA samples are significantly different from PBS also? I guess it does, but I think it would be more convenient to state it clearly. The significance vs the PBS is the crucial point here to demonstrate the effect of NB-EPA.

Fig 4 D and E: both Akt phosphorylation and Ki67/DCX expression is increased in both PBS and NB-EPA with respect to the control. Can the authors provide an explanation of why these parameters seem to have the same pattern in untreated (PBS) HI and in NB-EPA treated mice?

Fig 6B and E: why do the western blotting miss the control +PTFalfa or +shPUMA, which are present in fig 6A, C and D?

Reviewer 2 Report

The authors identified N-benzyl eicosapentaenamide (NB-EPA) in Maca as a new kind of macamide and showed the protective property of NB-EPA against ischemic neuronal damage in vivo and in vitro. The major concern is as follows: how the authors determined the dose of NB-EPA (250µg per mouse)? Because NB-EPA was newly identified in this paper, the authors need to perform dose-response experiments using HIE mouse model.

Result section:

  1. Figure 4C & D: The phosphorylation of Akt increased in both PBS- and NB-EPA-treated groups. The authors explained that NB-EPA improved neuronal cell survival and proliferation through activation of phosphorylated AKT signaling which is OK. However, the data on the increase in Akt phosphorylation of neonatal mice after HI brain injury were not discussed.
  2. Figure 4E: Same as above.
  3. The purpose of doublecortin- and Ki-67-immunostaining should be explained.

Minor points:

  1. The authors use PBS as a vehicle. However, NB-EPA seems to be very hydrophobic. The authors really dissolved NB-EPA in PBS?
  2. “Neun” should be spelled ”NeuN”.
  3. An abbreviation needs to be spelled out at the first instance of a term in the abstract, again in the main text, and used consistently thereafter.

Reviewer 3 Report

The manuscript presents interesting results and is written in a very accessible way. However, I have few comments and questions.

  1. Numerical data should be included in included in the Results section
  2. 3D – intriguingly small number of apoptotic cells on pictures and quite big number on the graph. Please explain.
  3. Why DG and CA3 regions of the hippocampus were analysed (Fig. 3 and 4). It is well known that DG and CA3 are rather resistant to HI and the most damaged region is CA1. Additionally, the picture presented on Fig. 3A, panel PBS – presented sections seem more to be destroyed during tissue processing then present damage resulted from HI.
  4. I have also doubts concerning the age of animals subjected to behavioral tests. Description in methods suggests that tests were performed by 2 weeks old mice. Is it really the appropriate age for used tests?

Round 2

Reviewer 1 Report

This reviewer would like to congratulate authors for their paper

Author Response

We appreciate the generous encouragement from the reviewer.

Reviewer 2 Report

Minor points:

Response 1: In this study, the preparation of NB-EPA was divided into three steps. First, it was dissolved with DMSO, then diluted with PBS, and finally dissolved with ultrasound. Furthermore, it was dissolved by ultrasound again before each treatment. In all these experiments, the final concentration of DMSO was less than 0.3%. Hence, it contains the same concentration of DMSO in vehicle PBS group.

Recommendation: In this case, Vehicle is suitable instead of PBS. Please correct PBS to Vehicle throughout the manuscript.

Author Response

In this study, the preparation of NB-EPA was divided into three steps. First, it was dissolved with DMSO, then diluted with PBS, and finally dissolved with ultrasound. Furthermore, it was dissolved by ultrasound again before each treatment. In all these experiments, the final concentration of DMSO was less than 0.3%. Hence, it contains the same concentration of DMSO in vehicle PBS group.

Point 1: Recommendation: In this case, Vehicle is suitable instead of PBS. Please correct PBS to Vehicle throughout the manuscript.

Response 1: We carefully checked and changed “PBS” to “Vehicle” in the revised manuscript.

Reviewer 3 Report

The authors improved the manuscript according to the reviewer's comments. The manuscript is well written and presented data interesting. However, I still have some doubts concerning histological data. The authors’ explanation why the DG and CA3 region of hippocampus was presented is good but to dispel doubts the NB-EPA effect on CA1 region should be also presented.

Minor comments:

Page 3, line 88 – I would change „attenuates” into “reduces”

Page 3, line 91 – change “edema” into “swollen”

How was the dose of NB-EPA established?

Author Response

This manuscript is a resubmission of an earlier submission. The following is a list of the peer review reports and author responses from that submission.